# Genome-Wide Association Study of Starch Properties in Local Thai Rice

**DOI:** 10.3390/plants12183290

**Published:** 2023-09-17

**Authors:** Parama Praphasanobol, Putut Rakhmad Purnama, Supaporn Junbuathong, Somsong Chotechuen, Peerapon Moung-Ngam, Waraluk Kasettranan, Chanita Paliyavuth, Luca Comai, Monnat Pongpanich, Teerapong Buaboocha, Supachitra Chadchawan

**Affiliations:** 1Biological Sciences Program, Faculty of Science, Chulalongkorn University, Bangkok 10330, Thailand; p.praphasanobol@gmail.com; 2Center of Excellence in Environment and Plant Physiology, Department of Botany, Faculty of Science, Chulalongkorn University, Bangkok 10330, Thailand; pututrakhmad@gmail.com; 3Bioinformatics and Computational Biology Program, Graduate School, Chulalongkorn University, Bangkok 10330, Thailand; 4Pathum Thani Rice Research Center, Ministry of Agriculture and Cooperatives, Thanyaburi, Pathum Thani 12110, Thailand; junbuathongs16@hotmail.com (S.J.); pan_ku63@hotmail.com (P.M.-N.); 5Division of Rice Research and Development, Rice Department, Ministry of Agriculture and Cooperatives, Bangkok 10900, Thailand; somsongch@hotmail.com; 6Department of Botany, Faculty of Science, Chulalongkorn University, Bangkok 10330, Thailand; waraluk.k@chula.ac.th (W.K.); chanita.p@chula.ac.th (C.P.); 7Department of Plant Biology and Genome Center, University of California Davis, Davis, CA 95616, USA; lcomai@ucdavis.edu; 8Department of Mathematics and Computer Science, Faculty of Science, Chulalongkorn University, Bangkok 10330, Thailand; monnat.p@chula.ac.th; 9Center of Excellence in Molecular Crop, Department of Biochemistry, Faculty of Science, Chulalongkorn University, Bangkok 10330, Thailand; teerapong.b@chula.ac.th; 10Omics Sciences and Bioinformatics Center, Faculty of Science, Chulalongkorn University, Bangkok 10330, Thailand

**Keywords:** apparent amylose content, GBSSI, genome-wide association study, granule-bound starch synthase, resistant starch, sodium/calcium exchanger protein

## Abstract

Rice (*Oryza sativa* L.) is the main source of energy for humans and a staple food of high cultural significance for much of the world’s population. Rice with highly resistant starch (RS) is beneficial for health and can reduce the risk of disease, especially type II diabetes. The identification of loci affecting starch properties will facilitate breeding of high-quality and health-supportive rice. A genome-wide association study (GWAS) of 230 rice cultivars was used to identify candidate loci affecting starch properties. The apparent amylose content (AAC) among rice cultivars ranged from 7.04 to 33.06%, and the AAC was positively correlated with RS (R^2^ = 0.94) and negatively correlated with rapidly available glucose (RAG) (R^2^ = −0.73). Three loci responsible for starch properties were detected on chromosomes 1, 6, and 11. On chromosome 6, the most significant SNP corresponded to *LOC_Os06g04200* which encodes granule-bound starch synthase I (GBSSI) or starch synthase. Two novel loci associated with starch traits were *LOC_Os01g65810* and *LOC_Os11g01580,* which encode an unknown protein and a sodium/calcium exchanger, respectively. The markers associated with GBSSI and *LOC_Os11g01580* were tested in two independent sets of rice populations to confirm their effect on starch properties. The identification of genes associated with starch traits will further the understanding of the molecular mechanisms affecting starch in rice and may be useful in the selection of rice varieties with improved starch.

## 1. Introduction

Increasing economic and social development during the 20th century has increased aging in the world’s population. This trend will continue [1]. According to the International Diabetic Federation, changes in human lifestyle and dietary preferences have led to an increase in health problems and diseases, as exemplified by the rising prevalence of obesity, type II diabetes, and coronary artery disease [2,3]. Dietary choices are thought to be important for maintaining health. Rice (*Oryza sativa* L.) is one of the most important crops in the world and a main source of energy for humans, especially in Asia [4]. Starch, the main component of the grain’s endosperm, is the major source of carbohydrates in staple foods. It consists of two types of molecules, amylose and amylopectin [5], whose fractions vary depending on the source of starch. According to the International Rice Research Institute, rice starch can be classified into four groups based on the apparent amylose content (AAC): low or waxy (0–5%), low (5–20%), medium (20–25%), and high (26–33%) AAC [6]. AAC is the main starch factor affecting digestion and absorption in the small intestine. Additionally, starch can be categorized according to its digestion properties by the content of rapidly available glucose (RAG), slowly available glucose (SAG), and resistant starch (RS) [7,8]. RAG refers to glucose available for absorption in the small intestine within 20 min. Eating rice with a high RAG can lead to a rapid increase in blood glucose. SAG refers to glucose that becomes available for absorption in the small intestine within 120 min; rice with high SAG is digested slowly in the small intestine and results in a more gradual release of glucose in the blood. RS refers to the starch that cannot be digested in the small intestine but is beneficial for gut microbes in the large intestine. Moreover, RS can increase the feeling of fullness and decrease calorie intake, which may help reduce the risk of high blood glucose levels [9]. The glycemic index (GI) is an indicator used to measure the effect of carbohydrates on blood glucose levels after digestion. The glycemic response is directly related to digestible starch [10]. Rapidly digestible starch can increase blood glucose levels. Consumption of foods with low GI can reduce the prevalence of type II diabetes, obesity, hyperlipidemia, and cardiovascular disease [11]. 

Starch is synthesized through the coordinated activities of multiple enzymes, including ADP-glucose pyrophosphorylase (AGPase), starch synthase (SS), and the starch branching enzyme (SBE) [12]. AGPase is the key regulatory enzyme in starch biosynthesis, controlling starch levels in all plants [13,14]. *SS* can be divided into granule-bound starch synthase (GBSS) or *Waxy* (Wx), which regulates starch accumulation and is the main enzyme for the synthesis and elongation of amylose. Soluble starch synthase (SSS) is the main enzyme for amylopectin elongation [15,16]. Additionally, SBE is mainly involved in the structure of starch by catalyzing and creating a branch in glucose polymers [17]. 

A genome-wide association study (GWAS) is an effective method for predicting candidate genes based on the relationship between genetic and phenotypic variants. Presently, many genes are known to be responsible for rice starch properties. *GBSSI* controls the amount of amylose content [18,19,20] and is associated with the RS content in rice [9,21]. Additionally, it is the major gene responsible for the GI in rice [22,23,24]. Moreover, ADP-glucose pyrophosphorylase small subunit 1 (AGPS1)*,* starch synthase IIa (SSIIa), and isoamylase 1 (ISA1) are also associated with the RS content [9]. However, there is limited knowledge of genetic variants explaining the apparent amylose and digestible starch contents in Thai rice. In this study, we aim to identify candidate genes responsible for the starch property in Thai landrace rice varieties using GWAS and to develop molecular markers for starch traits in a rice breeding program. 

## 2. Results

### 2.1. Evaluation of Phenotypic Parameters

The value of the starch property (AAC: apparent amylose, RAG: rapidly available glucose, SAG: slowly available glucose, RS: resistant starch, HI: hydrolysis index, and GI: glycemic index) was evaluated in 230 rice cultivars (Appendix A). The AAC values ranged from 7.04% (Gam Meuang Nan) to 33.06% (Lai Mahk), with an average of 22.17 ± 0.82%. The RAG values ranged from 15.14% (Leb Nok) to 23.05% (Dam Dahng and E-Meud), with an average of 18.97 ± 0.71%. The SAG values ranged from 7.81% (Khao’ Niaw Dam Maw) to 14.44% (Leuang Bahn Sang), with an average of 11.14 ± 0.84%. RS values ranged from 2.11% (Khao’ Gam GS.23113) to 4.50% (Ta Nod), with an average of 3.14 ± 0.20%. Moreover, the HI and GI values ranged from 28.11% (Daw Yuan) to 46.82% (Hawm Nin) and from 55.53% to 65.42%, respectively. These values display significant correlations as shown in Figure 1. The AAC and RS values showed a strong positive correlation but were negatively correlated with the RAG. Additionally, no significant correlation was found in the SAG, the HI, and the GI (Figure 1, Appendix A).

Based on clustering analysis of starch properties, the 230 rice cultivars were separated into five groups (Appendix A). Group 1 consisted of 43 cultivars with a low AAC and RS, a medium SAG, a high RAG and a medium to high GI. Group 2 included 16 cultivars of low AAC and RS, a medium to high SAG, a medium to high RAG, and a low GI. Most rice members of groups 1 and 2 were sticky rice or waxy rice. Thus, groups 1 and 2 were divided from the other groups, which have higher values of AAC and RS. Group 3 contained 33 cultivars with low to medium AAC, RS, and SAG and with very high RAG and GI. Group 4 consisted of 88 cultivars with medium to high AAC, RS, and SAG and with medium RAG and GI. The last group (group 5) included 50 cultivars with medium to high AAC, RS, and SAG and with low RAG and GI (Figure 2, Appendix A). Among these groups, rice varieties in group 5 were the best candidates for the development of healthy food because of their low GI and high RS. These cultivars include ‘Hawm Gulahb Daeng’, ‘Nam Dawk Mai 595′, ‘Sai Bua 59-106-2′, ‘Khao Nahng Mon’, ‘Hawm Mali Daeng’, ‘Nahng Chalawng’, and ‘Leuang Bahn Sang’.

### 2.2. Genome Wide Association of Starch Traits in Thai Rice Populations 

These cultivars were used for entire genome sequencing. The SNPs were distributed across the genome. The density of the SNPs in 12 chromosomes of *Oryza sativa* is shown in Figure 3. In total, 346,448 high-quality SNPs were genotyped across the rice population.

#### 2.2.1. Association Mapping

To identify the genetic loci responsible for the variation in the starch properties of the 230 rice cultivars, GWAS was conducted based on genomic SNP, and Manhattan plots were generated to determine loci significantly associated with starch properties (Figure 4). Based on these GWAS, 156 SNPs located on 32 causative loci were identified (Figure 4, Table 1), suggesting the involvement of these genes in starch characteristics and the amylose content in rice. At least three significant peaks on the GWA mapping for AAC, RAG, and RS were detected on chromosomes 1, 6, and 11 with a threshold of −log(*p*) = 6.84. Although other SNPs occurred on chromosomes 2 and 7, their significance level was lower. No significant SNPs were determined by the association of SAG, HI, and GI (Figure 4). 

There were 15 loci that were consistently identified based on the association of three traits: AAC, RAG, and RS, while 5 loci were identified by the AAC trait only (Figure 5, Table 1). These findings demonstrate that the analysis of the AAC trait can reveal more causative loci. 

Based on the AAC trait, the causative loci with the most significant (the lowest *p* value) SNPs are on chromosomes 1, 6, and 11 (Figure 4A). On chromosome 1, the significant SNPs are in *LOC_Os01g65810*, which encodes a putative unknown protein, while on chromosome 6, the significant SNPs are in several loci (Table 1), with the most significant SNPs in *LOC_Os06g04200*, encoding starch synthase or granule-bound starch synthase I (GBSSI). Within this gene, the analysis detected a total of 14 SNPs located in 5′ UTR, exon 1, intron 9, and 3′end regions (Appendix A). On chromosome 11, the most significant SNPs were in *LOC_Os11g01580*, encoding a sodium/calcium exchanger protein. This is the first report associating an ion transporter to starch traits in rice. 

To investigate the linked segments, we determined pairwise linkage disequilibrium (LD) between the most significant AAC-associated SNPs on chromosomes 1, 6, and 11 (Figure 6). SNP100443 on chromosome 1 was associated with the AAC trait at the lowest *p* value (Figure 6A). There were five close LD blocks. LD blocks 1, 4, and 5 contained the uncharacterized gene (expressed proteins), while block 2, which is 16 kb long, contained the SNPs in *LOC_Os01g65780*, encoding the glycosyl transferase protein. LD block 3 contained two genes, *LOC_Os01g65790* and *LOC_Os01g65800*, encoding pectin esterase and the powdery mildew resistant protein 5, respectively (Figure 6B and Appendix A). Pairwise LD with SNP516807 on chromosome 6 demonstrated one large 33kb block and three smaller ones. *LOC_Os06g04200*, encoding starch synthase, is in all four blocks, with the most significant SNP in this region, SNP516807, located in block 1 (Figure 6C and Appendix A). SNP902089 is the most significant SNP on chromosome 11 (Figure 6A) and it is in block 6 as shown in Figure 6D and Appendix A. Block 1 contains the *LOC_Os11g01510* gene encoding the ubiquitin-activating enzyme, while *LOC_Os11g01530,* which encodes ferritin-1, lays over blocks 2 and 3. PMR5 (*LOC_Os11g01570*) is in block 5.

To demonstrate the association between alternative alleles in each locus with the AAC trait, the AAC and alleles found in 230 rice cultivars in each position of the causative genes, *LOC_Os01g65810*, *LOC_Os06g04200*, and *LOC_Os11g01580*, are shown in Figure 7. A single SNP was detected in *LOC_Os01g65810* (Figure 7A). The heterozygous allele (C/T) in Chr1:38218116 represents high AAC, while the homozygous allele (CC) represents low AAC (Figure 7A). On chromosome 6 (Figure 7B–O), SNPs in the 5′ UTR region, Chr6:1765761 (Figure 7B), Chr6:1765887 (Figure 7C), and Chr6:1766058 (Figure 7D) showed alternative alleles (different from reference genome), causing high AAC. These show that the alternative SNP in Chr6:1765761, the insertion at Chr6:1765887, and the deletion at Chr6:1766058 are associated with a high AAC. An insertion at position 1767006 in exon 1 of *LOC_Os06g04200* resulted in the highest significant SNP (−log(*p*) = 41.77) at this position (Figure 4A). Two different allelic patterns, G and GCCACGGGTTCCAGGGCCTCAA, were found. The insertion was well associated with low AAC (Figure 7E). Also, SNPs on chromosome 6 at position 1769158 (Figure 7F), 1769205 (Figure 7G), and 1769256 (Figure 7H) are in intron 9 of *LOC_Os06g04200*. The alternative alleles, which are the deletion in position Chr6:1769158, the SNP in position Chr6:1769205, and Chr6:1769256 led to an increase in AAC (Figure 7F–H). Moreover, the alternative alleles of SNPs in the 3′end region of *LOC_Os06g04200* located at positions 1770672 (Figure 7I), 1770763 (Figure 7J), 1770890 (Figure 7K), 1770909 (Figure 7L), 1770964 (Figure 7M), 1771095 (Figure 7M), and 1771117 (Figure 7O) resulted in higher AAC. In addition, many loci on chromosome 6, which were associated with starch properties, affected an expressed protein and were located close to GBSSI. These genes are *LOC_Os06g04195, LOC_Os06g04210, LOC_Os06g04220, LOC_Os06g04230,* and *LOC_Os06g04240* (Table 1). The other loci on chromosome 6, *LOC_Os06g04080* and *LOC_Os06g04169*, were also associated with AAC, RAG, and RS contents. *LOC_Os06g04080* and *LOC_Os06g04169* encoded glycosyl hydrolases and alpha/beta hydrolase, respectively, which catalyze the hydrolysis of starch. Furthermore, three SNPs on chromosome 11 which were positions 334302 (Figure 7P), 334530 (Figure 7Q), and 334582 (Figure 7R) were in exon 2 of *LOC_Os11g01580*, encoding the sodium/calcium exchanger protein. High AAC was associated with the heterozygous alleles in all positions, namely Chr11:334302 (Figure 7P), Chr11:334530 (Figure 7Q), and Chr11:334582 (Figure 7R), suggesting that the duplication in this gene had occurred and led to a high AAC.

#### 2.2.2. Molecular Marker Analysis

We designed markers to detect genetic variation in starch-controlling loci. Two molecular markers were developed to genotype loci affecting AAC, RAG, and RS in the rice population. The first molecular marker (marker I) was targeted to SNPs of exon 1 in *LOC_Os06g04200* (GBSSI). The second molecular marker (marker II) was developed to genotype *LOC_Os11g01580* (sodium/calcium exchanger protein). Two independent rice populations, each containing 100 cultivars, were used to study the correlation between the markers and the starch traits. The rice cultivar lists are shown in Appendix A. For the first molecular marker (marker I), the amplified PCR product was digested with restriction enzyme BpuEI. Two patterns were obtained. The first pattern consisted of double bands of 360 and 257 bp, and the second pattern consisted of three bands: 380, 360, and 257 bp (Appendix A). A higher % AAC, a lower % RAG, and a higher % RS associate with the double band pattern in both rice populations (Figure 8A–D). A second molecular marker (marker II) was developed to detect SNPs in *LOC_Os11g01580* (sodium/calcium exchanger protein). After amplification, two patterns, including a single band of 530 bp and triple bands of 570, 530, and 480 bp, were detected (Appendix A). The triple pattern was associated with higher % AAC and % RS and a lower % RAG (Figure 8E–H). 

Both markers can explain % AAC ranging from 0.50 to 0.85 R^2^. They were also correlated to the RS trait in both populations. Interestingly, for the RAG trait, both markers showed a much lower R^2^ in rice population I than in population II. It explained only 4–6% of the variation in population I, while it accounted for more than 30% of the RAG in population II. Neither marker could explain the SAG trait (Table 2). 

## 3. Discussion

### 3.1. Evaluation of Rice Phenotypes 

Thai rice varieties displayed a wide variation in the apparent amylose content (AAC). The AAC of 230 rice cultivars ranged from 7.04 to 33.06% (Appendix A). All cultivars are indica rice. Most of them are Thai landrace, collected from various parts of Thailand (Figure 9) and do not have genetic relationships. They were selected for the study based on the variation in AAC. Only some cultivars were genetically related: ‘Tab Tim Chumpae (RD 69)’ whose parental cultivars are ‘KDML105’ and ‘Sang Yod’ and ‘RD 43’ whose parents are ‘Supanburi 1’ and ‘Hawm Supanburi’. According to Juliano [6], the rice in this study can be classified into four groups, as follows: Rice with very low AAC (<10%) ranged from 7.04 to 9.06 % and contained mostly sticky rice or glutinous rice. Rice with low AAC (10–20%) showed AAC values ranging from 11.19 to 20.12%. Rice with medium AAC (20–25%) displayed AAC values from 21.16 to 25.98%, and rice with high AAC (>25%) ranged from 26.09 to 33.06% (Appendix A). However, in this study, by combining AAC and digestible starch, we demonstrated five clusters: Group 1 is low to medium amylose and resistant starch (RS), medium slow available glucose (SAG), high rapidly available glucose (RAG), and medium to high (GI); group 2 is low to medium amylose and RS, medium to high SAG, medium to high RAG, and low GI; group 3 is low to medium amylose, RS, and SAG and very high RAG and GI; group 4 is medium to high amylose, RS, and SAG, medium RAG and GI; and the last group (group 5) is medium to high amylose, RS, and SAG and low RAG and GI (Figure 2; Appendix A). ‘RD 69’ and their parental cultivars are clustered in group 1. This is consistent with the purpose of the breeding program of ‘RD 69’, i.e., to develop soft texture rice seeds with high antioxidants. Both ‘KDML105’ and ‘Sang Yod’ have low AAC, leading to the soft texture of the seeds and the high antioxidant traits, which were contributed from Sang Yod, the colored rice with high antioxidant properties. ‘RD 43’ is also clustered in group 1, but its parental cultivar, ‘Supanburi 1’, is clustered in group 4, which has medium to high amylose, RS, and SAG. Unfortunately, we did not have the phenotypes of ‘Hawm Supanburi’ in this study to discuss the possibility of a genetic contribution for ‘RD 43’ phenotypes. It may be assumed that the difference in AAC between ‘RD43’ and its parental cultivar ‘Supanburi 1’ came from ‘Hawm Supanburi’. 

When compared, the clustered group with the original location of the cultivars, group 1 and group 2, were mostly originally located in the north and northeastern part of Thailand. Most of group 4 and group 5, which contain a higher level of AAC, were originally from central Thailand (Figure 9). These are consistent with the consumption behavior of Thai people. Sticky rice (low AAC) is more popular in the north and northeastern part of Thailand, while it is less popular in the central part of Thailand.

Among these five groups, the rice in group 5 has the most desirable nutritional characteristics as indicated by the high RS and the low RAG/GI. The AAC was positively correlated with the RS content, suggesting that AAC may be the main factor determining the RS content in rice. Rice starch with a high AAC resulted in a high RS content in various rice varieties, consistent with the previous study [9,21,23,32]. Several studies found a correlation between the amylose content and the digestibility of starch [22,33,34]. Li et al. [33] showed that the indica subspecies with the highest AAC had a slower digestion rate than the japonica subspecies [34]. Anacleto et al. [22] found that the AAC had a weakly negative correlation with GI, but we could not detect this correlation in this rice population in our study. Zhang et al. [34] also found a negative correlation between AAC and RDS (slowly digestible starch). The RAG and RS were measured using in vitro starch digestion. The RAG reflected the rate of glucose available for absorption in the small intestine within 20 min, which is a key determinant of the glycemic response [7,35,36]. Foods with high AAC and RS and a low RAG have considerable health benefits due to the slower release of glucose and correspondingly lower glycemic response. Similarly, we observed a negative correlation between the AAC and the RAG and no correlation among the AAC/RS and the SAG and the GI (Figure 2, Appendix A).

### 3.2. Association Mapping and Molecular Marker Analysis

In the present study, a total of 18 SNP positions on chromosomes 1, 6, and 11 contained the highest significant loci responsible for the starch property (Appendix A). On chromosome 1, the SNP was in *LOC_Os01g65810*, which encodes an unknown protein. When compared to the reference japonica rice genome (*Oryza sativa* cv. *Nipponbare*), the selected indica rice cultivars in this study have SNPs and INDEL predicted to result in a truncated protein (Appendix A). This may be the cause of the different starch textures and properties between the japonica and the indica rice. Moreover, in the high AAC cultivar, the C to T SNP at Chr1:38218116 led to an increase in the AAC’s value (Figure 7A). The heterozygous allele, C/T, at this position, suggests that duplication of this region led to the higher % AAC. This SNP was also responsible for the RAG and RS variations in rice. Therefore, the allelic variation at this locus allowed the identification of allelic patterns associated with the different AAC groups of rice. On the LD block, there are many SNPs located on block 2 (Figure 6B, Appendix A), which included *LOC_Os01g65780* and *LOC_Os01g65790*; these two loci encode glycosyl transferase and pectin esterase, respectively. Although *LOC_Os01g65810* encoded an unidentified protein in rice, it was found near *LOC_Os01g65780*, which has a clear orthologous gene in Arabidopsis (AT3G18660) and encodes plant glycogenin-like starch initiation proteins (PGSIPs). PGSIPs are predicted to be chloroplast localized, and a knockout of PGSIP1 in Arabidopsis decreased starch accumulation in the leaves [37]. Moreover, *LOC_Os01g65790* encoded pectin esterase. Pectin is one of the main components of the plant cell wall, controls cell wall porosity, and constitutes the major adhesive material between cells [38]. In plants, pectin esterases are hydrolytic enzymes that play a role in cell wall metabolism. It is likened to *LOC_Os01g65810*, as it showed up in the high LD, indicating that they might be inherited together. This result suggests that this protein may play a role in starch biosynthesis and possibly amylose/amylopectin initiation in rice grains. 

We identified multiple loci related to starch properties on chromosome 6. As expected, the most significant SNPs were in *LOC_Os06g04200,* which encodes starch synthase or granule-bound starch synthase I (GBSSI). This locus affected AAC, RAG, and RS contents. Previous studies showed a significant correlation between SNPs in the intron 1 (G/T), in the coding regions exon 6 (A/C) and exon 10 (C/T), and the AAC of GBSSI [18,19,39]. However, the present study identified SNPs located in the 5′UTR, exon 1, intron 9, and 3′end regions (Appendix A). A major effect on starch properties may be caused by an INDEL in exon 1 (G/GCCACGGGTTCCAGGGCCTCAA) at position 1767006 bp (Figure 7E), which causes the change from proline codon (Pro) to stop codon (Appendix A). Thus, this suggests that the insertion in exon 1 results in a reduction in the GBSSI affecting the amount of amylose in the endosperm. Another SNP on the 5’ UTR (T/G) at position 1765761 bp had a strong effect (Figure 7B); the altered allele G resulted in an increase in the AAC value, which could explain rice AAC differences. This position was in the first LD block in this region (Figure 6C). This LD block includes other linked loci, namely *LOC_Os06g04150*, *LOC_Os06g04169*, and *LOC_Os06g04190* (Appendix A). Previous studies identified the GBSSI as the enzyme responsible for AAC variation in rice. The GBSSI controlled the amount of RS and was correlated with the AAC content in rice grains [9,22,25,26,27,28,29,30]. This result is supported by the fact that RS was positively correlated with the AAC. In addition, the AAC was also negatively correlated with the RAG; therefore, it is possible that the RAG is also controlled by the GBSSI. Other loci on chromosome 6, *LOC_Os06g04080* encoding glycosyl hydrolases and *LOC_Os06g04169* encoding alpha/beta hydrolase, were also associated with AAC, RAG, and RS contents (Table 1). Glycosyl hydrolases and alpha/beta hydrolases hydrolyze the glycosidic bond in polysaccharides of the cell wall, likely affecting cell wall architecture [40]. This finding is reinforced by the fact that many SNPs are located on block 1, which included *LOC_Os06g04169*, and they are near *LOC_Os06g04200*, which is in high LD, indicating that they are most likely to be inherited together (Figure 6C, Appendix A). Although there is no evidence that hydrolase enzymes play a direct role in starch biosynthesis, these enzymes affect starch hydrolysis. Taken together, it is possible that these enzymes may influence starch properties by affecting digestible starch.

Furthermore, significant SNPs (*p* = 6.91 × 10^−17^) on chromosome 11 were in *LOC_Os11g01580*, which encoded the sodium/calcium exchanger protein (NCX). Interestingly, at this locus, the SNPs associated with % AAC are heterozygous (Figure 7P–R). Positions Chr11:334302 and Chr11:334530 show an INDEL allele, while Chr11:334582 shows an SNP allele. This suggests that the duplication of the *NCX* gene occurred in these haplotypes and that it may be related to the increase in % AAC. This is also supported by the pattern of the PCR-RFLP marker II, which was designed to detect the variation in this gene. This marker showed either a single amplified band or triple amplified bands (Figure 8E–H). In the previous study, the NCX protein in both Arabidopsis and rice was found to play a role in salt stress and controlling Ca^2+^ in cell [41,42]. In this study, they are not linked to other genomic locations nearby, and no specific gene has been conclusively implicated in starch properties (Figure 6D, Appendix A). Little is known about the involvement of the NCX protein in starch biosynthesis. Interestingly, no other candidate genes were discovered near *LOC_Os01g65810* and *LOC_Os11g01580*, suggesting that they represent novel candidates for the starch property in rice. The function of these genes in relation to starch properties should be investigated in the future.

Based on the results of marker analysis, rice in groups 1 and 2 was associated with triple bands of marker I and a single band of marker II, while rice in groups 4 and 5 tended to associate with double and triple bands of marker I and marker II, respectively (Appendix A).

New molecular markers were developed from the highest significant SNPs associated with starch property traits, which are in exon 1 of *LOC_Os06g04200* and exon 2 of *LOC_Os11g01580*. After screening the 100 rice cultivars in two independent populations (Appendix A) with these molecular markers, they could be used to discriminate the rice variation between low and high AACs; it showed a high correlation (Figure 8). Accordingly, the AAC showed a positive correlation with the RS while the RAG showed a negative correlation with the AAC and the RS. Thus, these markers from this study provide efficient markers for predicting the starch property based on the AAC, the RAG, and the RS contents of rice.

## 4. Materials and Methods

### 4.1. Plant Materials and DNA Extraction

The rice population used in this study comprised 230 rice varieties (Appendix A) grown at the Pathum Thani Rice Research Center, Rice Department, Ministry of Agriculture. The experiment used a completely randomized design with five replicates. This experiment was conducted during in-season rice cultivation from August to December 2017. DNA was extracted from young leaf tissue using the Genomic DNA Mini Kit (Plant; Geneaid Biotech Ltd., Taiwan) according to the manufacturer’s protocol. Genomic DNA (gDNA) was quantified using a Nanodrop^TM^ spectrophotometer and confirmed using agarose gel electrophoresis. Genomic DNA was sequenced using a Genome Analyzer (Illumina, Inc., San Diego, CA, USA).

### 4.2. Evaluation of Phenotypic Parameters

#### 4.2.1. Apparent Amylose Content (AAC)

The rice samples were dehulled into brown rice. Subsequently, they were ground and sieved through a 100-mesh sieve to measure the AAC, which was determined colorimetrically, measuring the amylose–iodine complex according to the modified method of Juliano [43]. A total of 20 mg of rice powder was weighed, placed in a 25 mL volumetric flask, and dispersed by adding 200 µL of 95% ethanol. Further, 1.8 mL of 1 M NaOH was added and incubated at 100 °C for 10 min. An aliquot (1 mL) was collected and mixed with 200 µL of 1M acetic acid and 400 µL of iodine solution. Distilled water was added to a final volume of 20 mL. The mixture was then incubated in the dark for 20 min. The absorbance of each sample was measured at 620 nm against a reagent blank using a microplate reader (SpectraMax^®^M3, San Jose, CA, USA). The percentage of apparent amylose content was calculated from the absorbance using the standard curve of amylose from potatoes (Sigma-Aldrich, Co., Darmstadt, Germany).

#### 4.2.2. Digestible Starch Measurement

Analyses of digestible starch, including RAG, SAG, total glucose (TG), and RS were performed according to the modified method of Englyst et al. [7]. Thirty milligrams of milled brown rice were weighed into two Eppendorf tubes. Each tube was digested with 1 mL of an enzyme solution consisting of 40 U/mL pancreatin (Sigma–Aldrich, Co., Darmstadt, Germany) and 3 U/mL amyloglucosidase (AMG) (Megazyme, Bray, Ireland), and two glass beads (5 mm diameter) were added to an Eppendorf tube. Both tubes were incubated at 37 °C with shaking at 250 rpm. After 20 min, an aliquot of 0.1 mL was taken from the first tube and mixed with 900 µL of absolute ethanol to end enzymatic hydrolysis for the determination of RAG, and for the second tube, the enzyme activity was ended using the same protocol after 120 min for SAG determination. For the remaining solution from the second tube (0.8 mL), the reaction was stopped using boiling in water at 100 °C for 30 min and then cooling on ice for 10 min. Subsequently, 430 µL of 7 M KOH was added and incubated on ice for 30 min. To a 0.2 mL aliquot, 2000 µL of 0.5 M acetic acid and 5.5 µL of AMG (3260 U/mL) were added, and the mixture was incubated at 70 °C for 30 min. The solution was placed in a boiling water bath for 10 min and then cooled to room temperature to obtain the TG. Glucose oxidase peroxidase or GOPOD reagent (Megazyme, Bray, Ireland) was used to estimate the glucose concentration at 20 min (G_20_), 120 min (G_120_), and TG. Aliquots of 50 µL of each solution were incubated with 1.5 mL of GOPOD at 50 °C for 10 min. The absorbance was read at 510 nm against a reagent blank. The starch fractions were calculated as follows:RAG = G_20_,(1)
SAG = G_120_ − G_20_,(2)
RS = (TG − G_120_) × 0.9.(3)

#### 4.2.3. Glycemic Index 

In vitro methods were used to measure the HI and the GI. The protocol was based on a modified method described by Kumar et al. [44]. A total of 20 mg of rice powder was added to distilled water and boiled in water at 100 °C for 2 min. The solution was incubated with 5 mL of 0.1 M phosphate buffer (pH 6.9) and 200 µL of pepsin (250 mg/mL), adjusting the pH to 2.5 with 10% *o*-phosphoric acid at 37 °C and shaking at 200 rpm for 60 min. The solution was mixed with 200 µL of α-amylase (125 mg/mL) before being transferred to a dialysis tube (Spectra/Por^®^ Dialysis Tubing, 12–14 kD MWCO, Repligen, Waltham, MA, USA). The dialysis tube was placed in a beaker containing 40 mL of 0.1 M phosphate buffer (pH 6.9) and incubated at 37 °C at 150 rpm for 240 min. Aliquots (0.4 mL) were collected from each sample after 30, 60, 120, 180, and 240 min. A total of 1.2 mL of 0.4 M sodium acetate buffer (pH 4.75) was added to each aliquot, and 24 µL of AMG (3260 U/mL) was added to hydrolyze the digested starch and incubated at 50 °C for 30 min. Duplicate aliquots of 150 µL each of timing solution were incubated with glucose oxidase peroxidase reagent at 50 °C for 20 min, and the absorbance was measured at 510 nm using a microplate reader. Maltose was used as the standard carbohydrate. The HI was calculated by dividing the area under the curve of each sample by that of maltose. HI was converted into a glycemic index using the formula given by Goni et al., as shown in Equation (4) [45]:GI = 39.71 + (0.549 × HI).(4)

### 4.3. Analysis of Genome-Wide Association and Molecular Markers 

#### 4.3.1. Association Mapping and Candidate Gene Analysis

Short reads obtained from the Illumina Genome Analyzer (Illumina, San Diego, CA, USA) were demultiplexed and mapped onto the rice reference genome sequence downloaded from the Plant Ensemble database (version IRGSP-1.0) (https://plants.ensembl.org/index.html, accessed on 8 February 2020) using the Burrows–Wheeler Aligner (BWA version 0.7.1). Variants were called using the Genome Analysis Toolkit (GATK; version 4.1). Variants were filtered using the following criteria: minimum coverage was 6 or 3 if positions with the minimum coverage of 6 were found in at least 10 accessions for a position to be called homozygous; and minimum coverage and minimum percentage of each of the two observed major base calls were 5 and 20, respectively, and minimum total coverage was 10 for a position to be called heterozygous. Identification of candidate genes underlying the genetic regulation of phenotypic traits using genome-wide association (GWA) mapping was performed using GEMMA Software (https://github.com/genetics-statistics/GEMMA) [1] based on the SNP (single nucleotide polymorphism) data and the phenotypic data. We selected SNPs in promoter and exon regions to perform association. SNPs were removed from the analysis by PLINK 1.07 [2] if they were not found in all accessions or their minor allele frequency (MAF) was less than 5%. To identify significant SNP, this study used the Bonferroni correction, where the significant threshold at 5% level was divided by the total number of SNP positions. The threshold level to declare an SNP as associated was −log(p) = 6.84. The association results were visualized using the quantile–quantile (Q–Q) plots and Manhattan plots using the CMplot packages. The MSU Rice Genome Annotation Project (http://rice.plantbiology.msu.edu) was used to search for the corresponding gene to each SNP. The linkage disequilibrium (LD) blocks were used to identify candidate gene regions using the Haploview software 4.2 [30]. The SNPs with highly significant associations in the block were identified and used as a measure of candidate genes between each locus because of their preference in association studies.

#### 4.3.2. Molecular Marker Analysis

The genotyping data for at least ten rice cultivars, including five rice cultivars with a low AAC and five with a high AAC (Appendix A), were used to develop two molecular markers from significant SNPs related to starch properties, which were visualized using the IGV software 2.3 to select the region that showed the most difference in expected sizes of amplified fragments for marker development. The primer was designed in the homozygous regions in all cultivars, and INDEL were located between the forward and reverse primers, when compared with the sequence of the low AAC and high AAC rice cultivars. Primers to detect the significant SNP on chromosome 1 could not be developed because it was a single SNP, which could not be detected by this method. Primers were designed using Primer blast (https://www.ncbi.nlm.nih.gov/tools/primer-blast/), based on GWAS results. Marker I was located on *LOC_O06g04200*; RSM1-F: 5′ AGCTTCAAATTCTAATCCCCAA, 3′ RSM2-R, 5′ TTACACATCCATCCAATGCG 3.’ Marker II was located on *LOC_Os11g01580*; RSM3-F: 5′ AACTTCGTCTCTGCTACGTT 3′, RSM4-R: 5′ CTCGTCG TGCTGTTCTAC 3′. The PCR was performed using a PCR reaction mixture of 20 µL containing 1 μg of DNA template, primers (0.5 μM each of forward and reverse primers), 0.5 μL of 10 mM dNTPs, 2 µL of 10X Taq buffer, 1 μL of 25 mM MgCl2, and 0.2 µL of Taq DNA polymerase (Thermo Fisher Scientific, Waltham, Waltham, USA). The PCR conditions were performed on a thermal cycler (Thermal cycler, Applied Biosystems, USA) as follows: an initial denaturation at 95 °C for 3 min; 34 cycles of denaturation at 95 °C for 30 s; annealing at 57.5 °C for primer 1 and 53.8 °C for primer 2 for 30 s and extension at 72 °C for 1 min; and a final extension at 72 °C for 5 min. The PCR product of primer set 1 (marker I) was digested with BpuEI enzyme (New England BioLabs, Ipswich, MA, USA) according to the manufacturer’s instructions. Next, the amplified PCR products of markers I and II were run on a 1% agarose gel and used to assess the band size with a 100 bp ladder marker (Enzynomics, Daejeon, South Korea). The designed markers were used to detect the AAC variations in two sets of 100 rice cultivars (Appendix A), each consisting of 50 rice cultivars with low AAC and 50 with high AAC. Regression analysis between the starch phenotype and electrophoretic banding patterns was performed using Excel 2021 software.

### 4.4. Statistical Analysis 

An analysis of variance was performed to detect differences among the means of each parameter in five replicates, and Tukey’s range test was used to detect significant differences between each mean at *p* < 0.05 using IBM SPSS Statistical Software version 22 (IBM Corp., Armonk, USA). Pearson correlation analysis was used to determine the relationships among the starch properties, and the correlation plots were constructed using the R “corrplot” package. All results are presented as mean ± standard error of the mean. Hierarchical cluster analysis using the Euclidean distance was used to draw relationships among rice cultivars based on the starch properties, and a dendrogram was generated based on Ward’s method using JMP Software version 9 (SAS Institute Inc., Cary, NC, USA). Box plots were created using the R ‘ggplot2′ package.

## 5. Conclusions

This study showed that the apparent amylose content displayed a strong positive correlation with the resistant starch and a negative correlation with the rapidly available glucose. Furthermore, the rapidly available glucose was negatively correlated with the resistant starch. Using a GWAS, three loci, namely *LOC_Os01g65810, LOC_Os06g04200,* and *LOC_Os11g01580,* contained the most significant SNPs and were identified as the determinants of starch properties in rice grains. These loci were used to develop molecular markers that could be used to predict starch properties and improve rice quality.

## 6. Patents

Chadchawan, S.; Buaboocha, T.; Pongpanich, M.; Praphasanobol, P. Rice Starch Marker 1 (RSM1) Rice Starch Marker 2 (RSM2). TH patent 2,203,000,528, 27 February 2022. 

Chadchawan, S.; Buaboocha, T.; Pongpanich, M.; Praphasanobol, P. Rice Starch Marker 3 (RSM3) Rice Starch Marker 4 (RSM4). TH patent 2,203,000,529, 27 February 2022.

## Figures and Tables

**Figure 1 plants-12-03290-f001:**
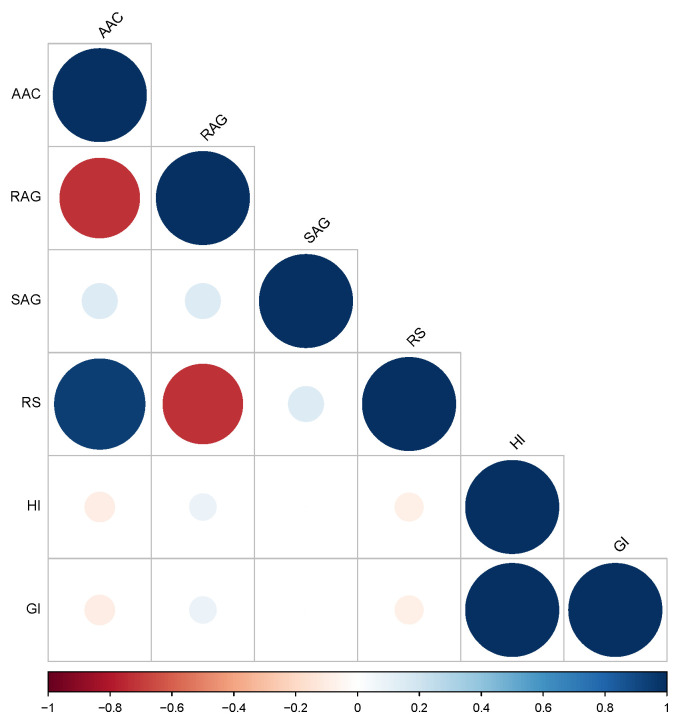
Pearson correlation of the starch property traits of 230 rice accessions. Positive correlations are displayed as blue circles and negative correlations as orange circles. The sizes of the circles are proportional to the correlation coefficients. Cells with insignificant correlation values (*p* < 0.05) are left blank.

**Figure 2 plants-12-03290-f002:**
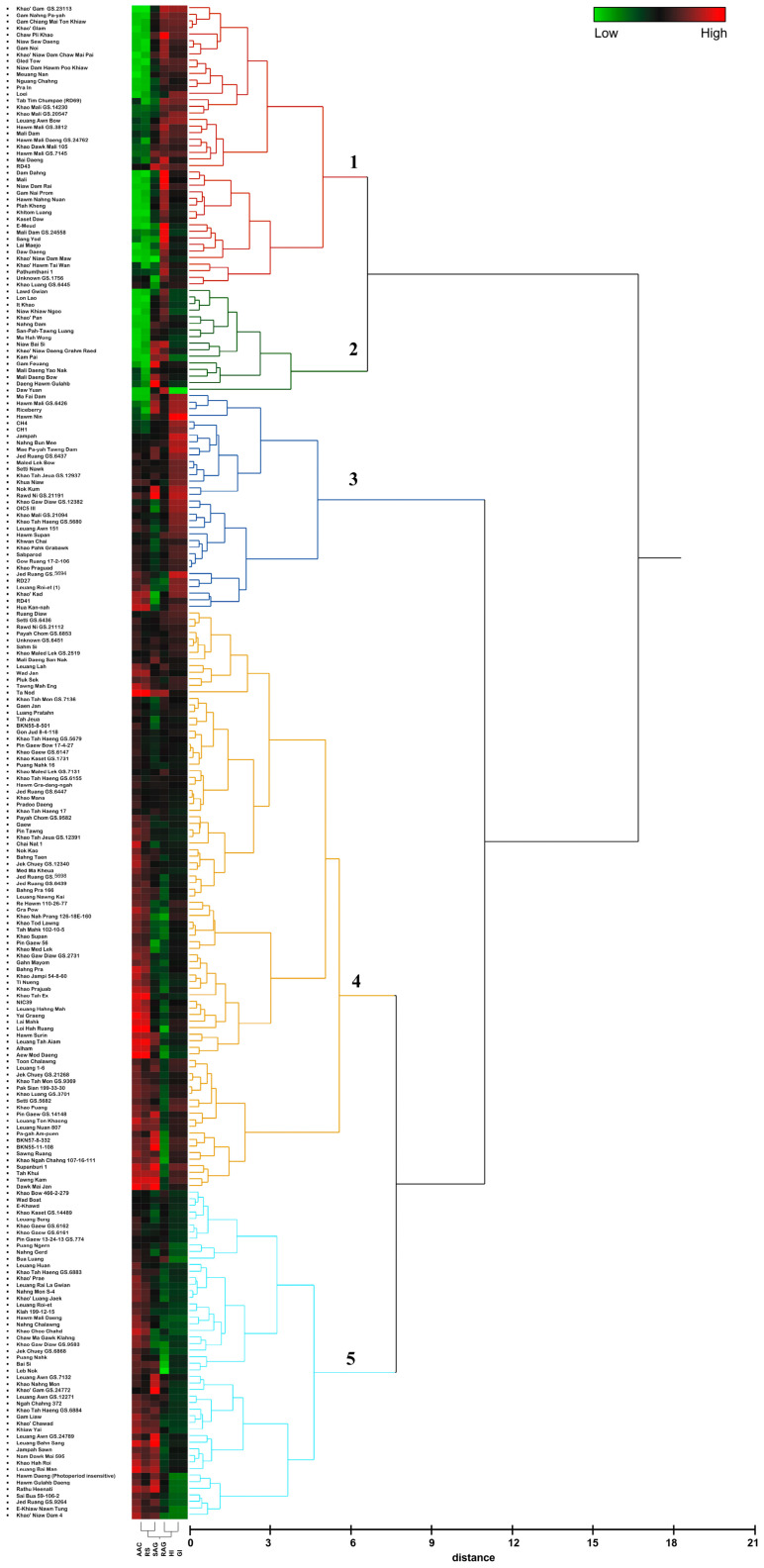
Hierarchical cluster analysis of the starch property in 230 rice accessions. Five cluster groups are shown with different color branches and the numbers represent the group numbers.

**Figure 3 plants-12-03290-f003:**
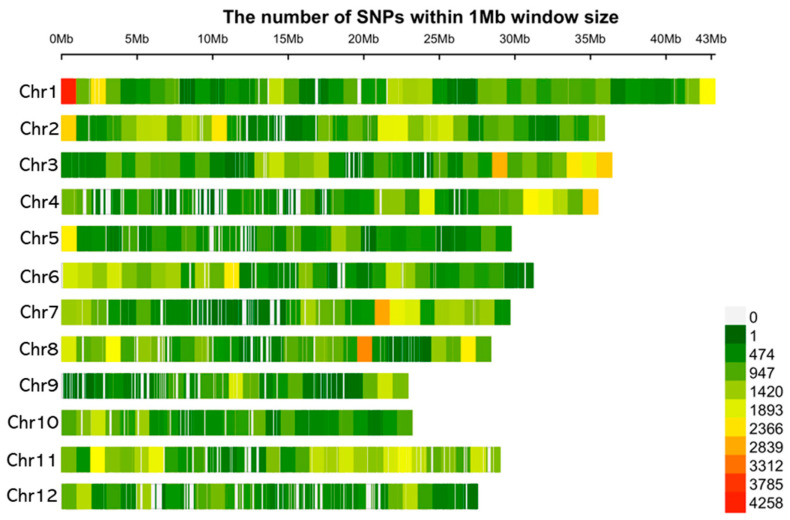
SNP density plot across the 12 chromosomes of rice representing the number of SNPs within a 1 Mb window size. The horizontal axis shows the chromosome length (Mb), while the different colors depict SNP densities. SNP: single nucleotide polymorphism.

**Figure 4 plants-12-03290-f004:**
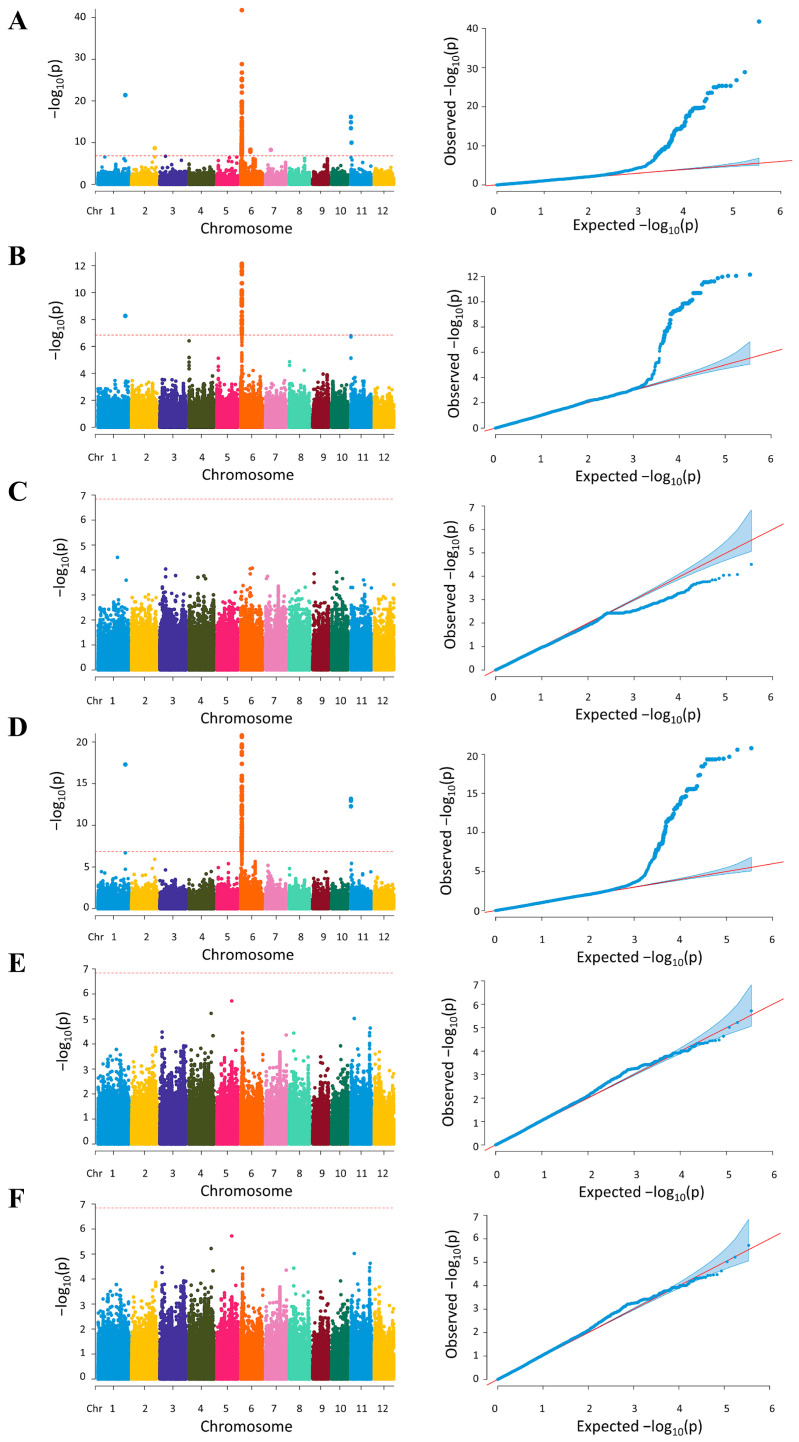
Manhattan plot (**left**) displaying the single nucleotide polymorphisms (SNPs) in a population of 230 rice cultivars associated with the following starch traits: (**A**) AAC, (**B**) RAG, (**C**) SAG, (**D**) RS, (**E**) HI, and (**F**) GI. The locations of SNPs on the chromosome and the association test (−log_10_(*p*)) are plotted on the x-axis and y-axis, respectively. The red line indicates the Bonferroni correction at *p* < 0.05. The chart (**right**) presents the quantile–quantile (Q–Q) plot of the observed and expected *p*-values from the association analyses.

**Figure 5 plants-12-03290-f005:**
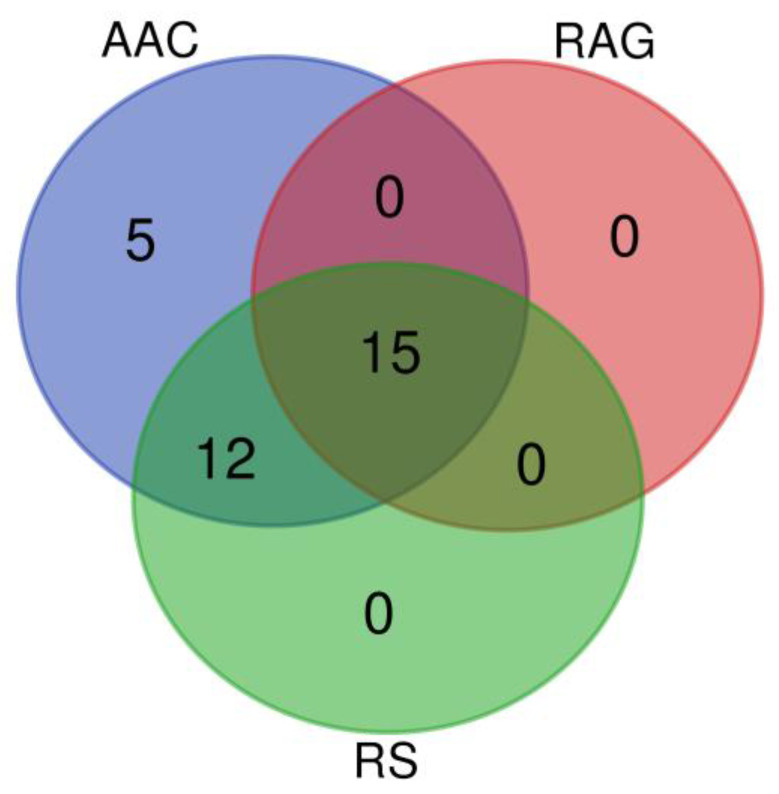
Venn diagram demonstrating the number of loci based on the SNPs associated with starch properties.

**Figure 6 plants-12-03290-f006:**
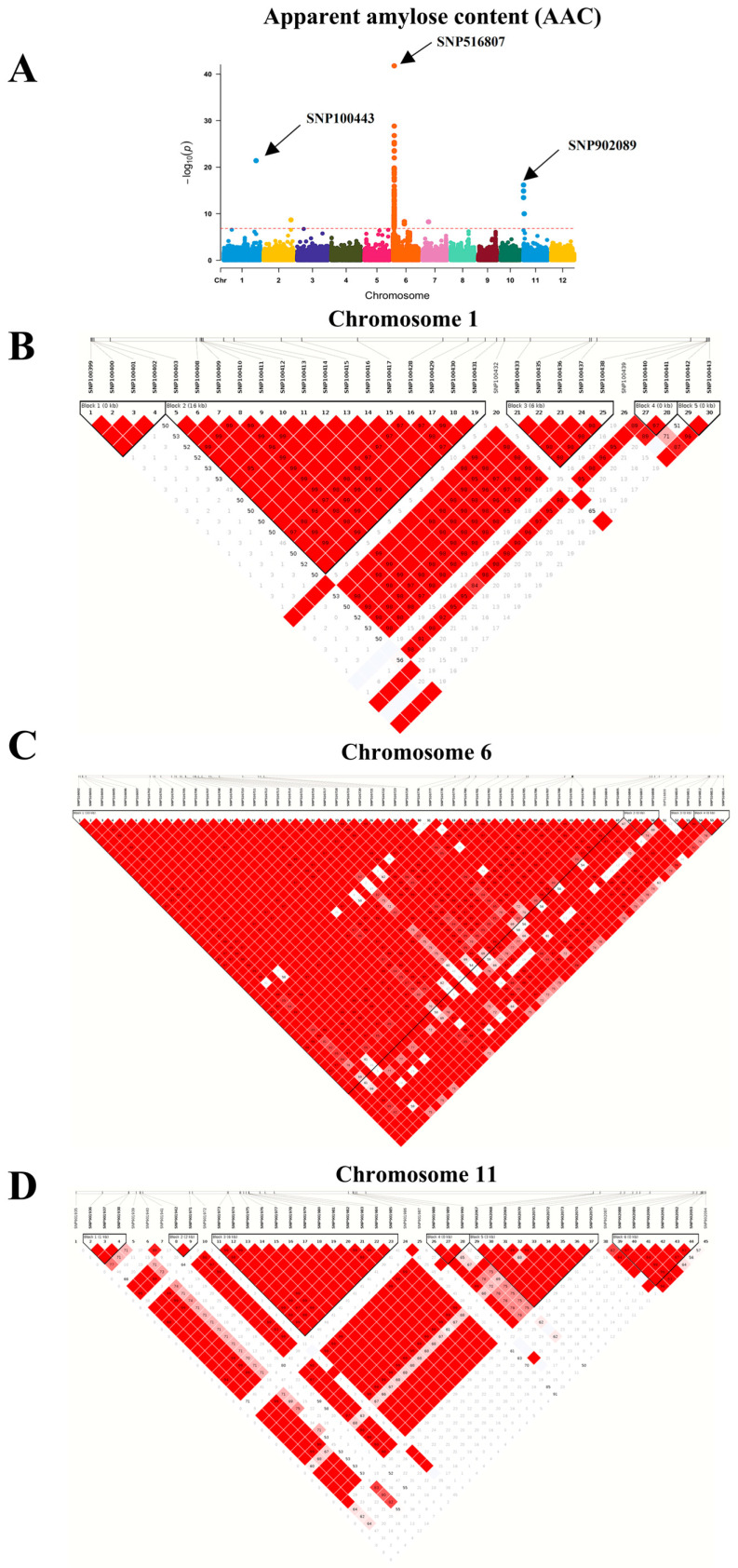
Manhattan plot of SNPs associated with the AAC trait with the most significant SNPs on chromosomes 1, 6, and 11 (**A**) and pairwise LD between the significant SNPs on each chromosome region (**B**–**D**), which are indicated as standard color scheme of Haploview [31], representing the values of D’/LOD. SNP: single nucleotide polymorphism, AAC: apparent amylose content; LD: linkage disequilibrium.

**Figure 7 plants-12-03290-f007:**
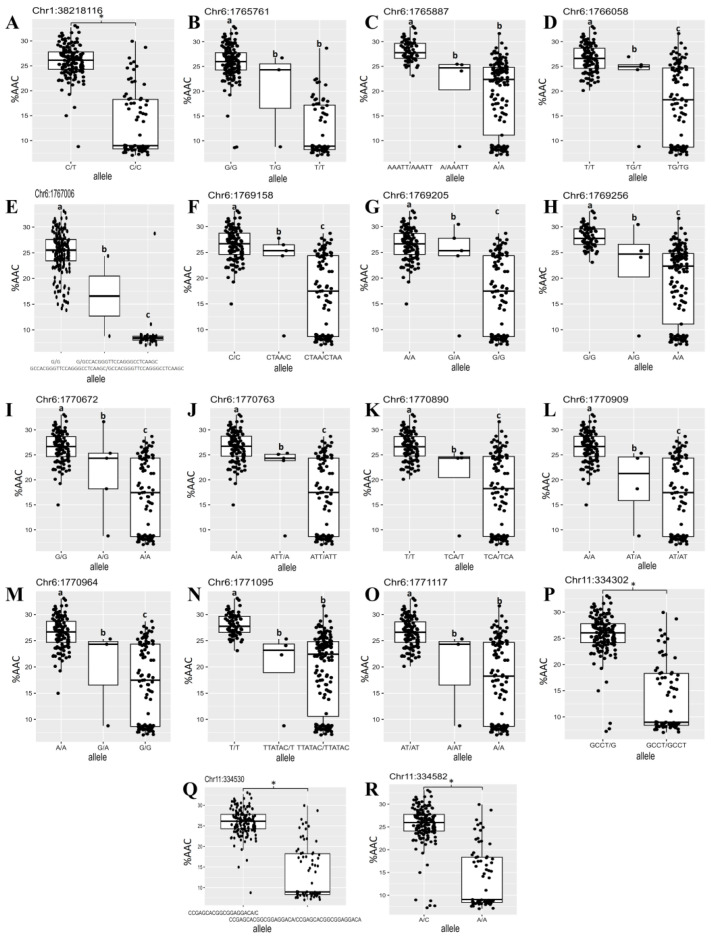
Boxplot of AAC based on the significance of the single nucleotide polymorphism of (**A**) Chr1: 38218116, (**B**) Chr6: 1765761, (**C**) Chr6: 1765887, (**D**) Chr6: 1766058, (**E**) Chr6: 1767006, (**F**) Chr6: 1769158, (**G**) Chr6: 1769205, (**H**) Chr6: 1769256, (**I**) Chr6: 1770672, (**J**) Chr6: 1770763, (**K**) Chr6: 1770890, (**L**) Chr6: 1770909, (**M**) Chr6: 1770964, (**N**) Chr6: 1771095, (**O**) Chr6: 1771117, (**P**) Chr11: 334302, (**Q**) Chr11: 334530, and (**R**) Chr11: 334582. The black horizontal lines represent the median values, the boxes represent the middle quartiles, and the whiskers are the range of data. Reference allele is on the rightmost column in each figure. * or the different letters above the bars represent the significant difference between the means of the alleles at *p* < 0.05 by DMRT or *t*-test, respectively.

**Figure 8 plants-12-03290-f008:**
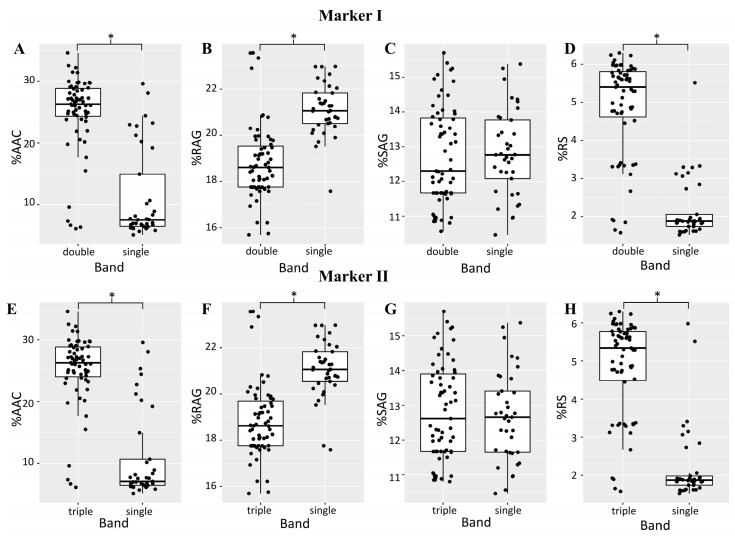
Box plots show the distribution of PCR-RFLP patterns of marker I (**A**–**D**) and marker II (**E**–**H**) and their starch properties, % AAC (**A**,**E**), % RAG (**B**,**F**), % SAG (**C**,**G**), and % RS (**D**,**H**). A hundred rice cultivars from population II (Appendix A) were used for this evaluation. * above the bars represents the significant difference between two PCR-RFLP patterns at *p* < 0.05 analyzed using a *t*-test.

**Figure 9 plants-12-03290-f009:**
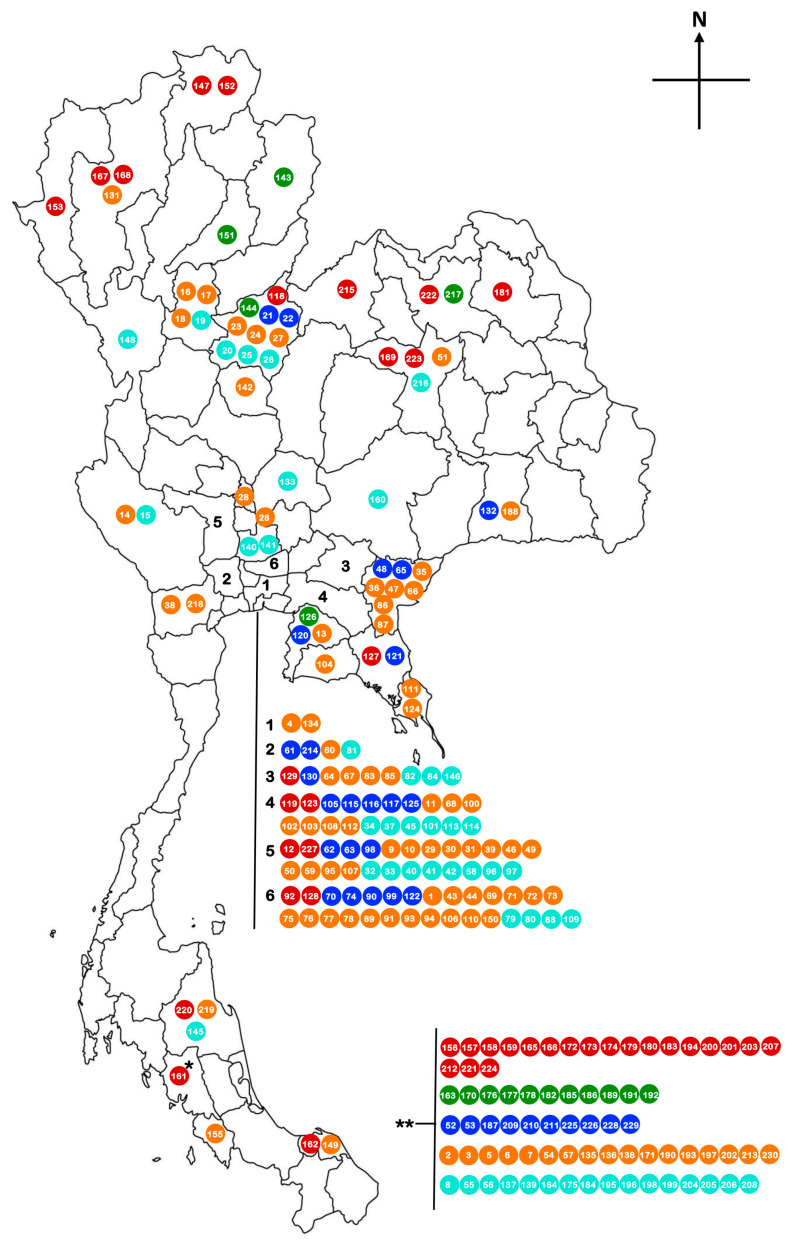
Distribution of rice cultivars in Thailand used in this study. The numbers of cultivars were indicated as shown in Appendix A. The color of each number represents the clustered groups in Figure 1, group 1: red, group 2: green, group 3: blue, group 4: orange, group 5: turquoise. * means that only the regional in the south of Thailand is known for this cultivar. ** means no information about the location of the cultivars.

**Table 1 plants-12-03290-t001:** List of genes identified by GWAS and *p*-value of the significant SNPs related to starch property traits.

Chr	Locus	Position	*p*-Value	Description	Parameter	Publication
**1**	*LOC_Os01g65810*	38218116	4.04 × 10^−22^	expressed protein	AAC, RAG, RS	-
2	*LOC_Os02g52950*	32380686	2.16 × 10^−9^	expressed protein	AAC	-
**6**	*LOC_Os06g03960*	1602299	7.05 × 10^−9^	expressed protein	AAC, RS	[25]
6	*LOC_Os06g03970*	1611971	3.55 × 10^−9^	receptor-like protein kinase 5 precursor	AAC, RS	[22,25]
6	*LOC_Os06g03980*	1616444	7.82 × 10^−10^	expressed protein	AAC, RS	[22,25]
6	*LOC_Os06g03990*	1629652	1.96 × 10^−10^	aminotransferase, classes I and II, domain-containing protein	AAC, RS	[22,25]
6	*LOC_Os06g04010*	1638628	1.85 × 10^−10^	GAGA-binding protein	AAC, RS	[22,25,26]
6	*LOC_Os06g04020*	1641473	1.96 × 10^−10^	histone H1	AAC, RS	[22,25]
6	*LOC_Os06g04030*	1644780	1.96 × 10^−10^	histone H3	AAC, RS	[22,25]
6	*LOC_Os06g04040*	1650499	1.96 × 10^−10^	WD domain, G-beta repeat domain-containing protein	AAC, RS	[22,25]
6	*LOC_Os06g04070*	1677364	2.97 × 10^−9^	pyridoxal-dependent decarboxylase protein	AAC, RS	[22]
6	*LOC_Os06g04080*	1691152	7.83 × 10^−10^	glycosyl hydrolases family 17, putative, expressed	AAC, RS	[22,26]
6	*LOC_Os06g04090*	1702532	8.59 × 10^−14^	no apical meristem protein	AAC, RS	[22]
6	*LOC_Os06g04130*	1730335	1.32 × 10^−15^	lung seven transmembrane domain-containing protein	AAC, RAG, RS	[22,25]
6	*LOC_Os06g04140*	1731629	2.30 × 10^−11^	expressed protein	AAC, RAG, RS	[22]
6	*LOC_Os06g04150*	1733997	4.88 × 10^−15^	magnesium-protoporphyrin O-methyltransferase	AAC, RAG, RS	[22,25]
6	*LOC_Os06g04169*	1740167	3.56 × 10^−15^	hydrolase, alpha/beta fold family domain-containing protein	AAC, RAG, RS	[22,25,26,27]
6	*LOC_Os06g04190*	1755645	1.89 × 10^−20^	rad1	AAC, RAG, RS	[22,25]
6	*LOC_Os06g04195*	1761270	1.44 × 10^−20^	expressed protein	AAC, RAG, RS	[25]
6	*LOC_Os06g04200*	1767006	1.69 × 10^−42^	starch synthase	AAC, RAG, RS	[22,25,26,27,28,29,30]
6	*LOC_Os06g04210*	1776682	3.89 × 10^−19^	expressed protein	AAC, RAG, RS	[28]
6	*LOC_Os06g04220*	1783512	4.81 × 10^−26^	expressed protein	AAC, RAG, RS	[22,26,28]
6	*LOC_Os06g04230*	1784113	1.63 × 10^−27^	expressed protein	AAC, RAG, RS	[22,25,28]
6	*LOC_Os06g04240*	1786344	2.83 × 10^−24^	expressed protein	AAC, RAG, RS	[22,25,28]
6	*LOC_Os06g04250*	1791200	2.29 × 10^−13^	phosphate-induced protein 1 conserved region domain-containing protein	AAC, RAG, RS	[22,28]
6	*LOC_Os06g04270*	1812542	9.66 × 10^−12^	transketolase, chloroplast precursor	AAC, RAG, RS	[25]
6	*LOC_Os06g04280*	1816427	8.21 × 10^−12^	3-phosphoshikimate 1-carboxyvinyltransferase, chloroplast precursor	AAC, RAG, RS	[22]
6	*LOC_Os06g04300*	1822237	3.86 × 10^−9^	tRNA 2-phosphotransferase 1	AAC	[22,25]
6	*LOC_Os06g23530*	13728922	4.94 × 10^−9^	pre-mRNA-splicing factor ATP-dependent RNA helicase	AAC	-
7	*LOC_Os07g12780*	7314685	5.56 × 10^−9^	cyclin	AAC	-
11	*LOC_Os11g01580*	334530	6.91 × 10^−17^	sodium/calcium exchanger protein	AAC, RS	-
11	*LOC_Os11g03130*	1129093	1.04 × 10^−10^	E2F-related protein	AAC	-

GWAS: Genome-wide association studies, SNP: single nucleotide polymorphism, AAC: apparent amylose content, RAG: rapidly available glucose, and RS: resistant starch.

**Table 2 plants-12-03290-t002:** Regression analysis for marker I (GBSSI) and marker II (*LOC_Os11g01580*) in 2 rice populations, each consisting of 100 cultivars.

	% AAC		% RAG		% SAG		% RS	
	R^2^	*p-*Value	R^2^	*p-*Value	R^2^	*p*-Value	R^2^	*p-*Value
Pop. I (Appendix A)								
Marker I	0.82	1.4 × 10^−38^	0.04	4.7 × 10^−2^	0.00	0.69	0.70	1.1 × 10^−26^
Marker II	0.85	2.8 × 10^−42^	0.06	1.6 × 10^−2^	0.00	0.96	0.81	1.3 × 10^−16^
Pop. II (Appendix A)								
Marker I	0.50	2.9 × 10^−16^	0.36	2.9 × 10^−11^	0.00	0.71	0.57	7.0 × 10^−20^
Marker II	0.54	2.5 × 10^−18^	0.34	2.0 × 10^−10^	0.00	0.60	0.55	1.5 × 10^−18^

## Data Availability

The data for this study are available from the corresponding author upon request.

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
