# Peer review of "Genome-Wide Association Study of Starch Properties in Local Thai Rice"

_plants, 2023, doi:10.3390/plants12183290_

Round 1

Reviewer 1 Report

plants-2535076

A manuscript entitled "Genome-wide association study of starch properties in local Thai rice" was reviewed for the Special Issue: Functional Genomics and Molecular Breeding of Crops of Plants.

 Correlation analysis among starch properties revealed that the amount of AAC positively correlates with RS, while it negatively correlates with RAG. These results well-fits with the previous findings. By GWAS using genome-wide SNPs and an amount of various starch species, three candidate loci were detected in Chrs. 1,6, and 11. PCR-RFLP analysis revealed that three candidate genes (GBSSI, Na/Ca exchanger, unknown protein) might be responsible for affecting starch properties.

Questions and comments

In General

Redundant descriptions of abbreviations for AAC, RG, RAG, SAG, HI, and GI were used in the Results and Figure legends of Fig.1,2,4,5,7, and 8. Since these abbreviations are explained in Introduction, I think it saves space to use only abbreviations in other parts.

The result of clustering whole accessions into 5 groups has not been well discussed in Discussion.

1)Are there any common characteristics within the same group? i.e., breeding history, the district where those samples were collected, or subspecies grouping (japonica, indica, or tropical japonica,), because generally, japonica has a relatively low amylose content while indica has a high amylose content.

2)The author developed the molecular markers to discriminate polymorphic mutations in GBSSI (marker I) or an ion transporter (marker II) with CAPS analysis. It seems that the two 100 rice varieties used in genotyping are a part of 230 rice varieties and that it would be interesting to overlap genotypes of GBSSI or a transporter gene onto results of Fig.2, to help elucidate the molecular mechanism of variation of starch property even partially.

Minor comments:

P2L93-95----“The value of the starch property (----) was evaluated among the 230 rice cultivars (Supplemented Table S1).” ----- TS1 shows only collective correlations among rice varieties and TS2 might be an appropriate table for this content.

P11L240,242,243 marker----to be a molecular marker or DNA marker.

P13L293,294---indica, japonica subspecies.

P14L368---In the previous study,

P15L416, P16L431---unify the description as “mixing at 200 rpm at 37°C for 60 min.”

P17L470---reward primer---reverse primer?

P18L539-----The authors express gratitude to Professor Luca Comai, but this makes a self-contradiction because he is one of the authors of this manuscript.

Author Response

Response to Reviewer I

Thank you very much for your comments. I would like to response to your comments as follows.

  1. Redundant descriptions of abbreviations for AAC, RG, RAG, SAG, HI, and GI were used in the Results and Figure legends of Fig.1,2,4,5,7, and 8. Since these abbreviations are explained in Introduction, I think it saves space to use only abbreviations in other parts.

Response

The abbreviations in the figure legend were deleted.

  1. The result of clustering whole accessions into 5 groups has not been well discussed in Discussion.

2.1 Are there any common characteristics within the same group? i.e., breeding history, the district where those samples were collected, or subspecies grouping (japonica, indica, or tropical japonica,), because generally, japonica has a relatively low amylose content while indica has a high amylose content.

Response

We added the discussion on common characteristics of some clustered groups and breeding history in the discussion.

2.2 The author developed the molecular markers to discriminate polymorphic mutations in GBSSI (marker I) or an ion transporter (marker II) with CAPS analysis. It seems that the two 100 rice varieties used in genotyping are a part of 230 rice varieties and that it would be interesting to overlap genotypes of GBSSI or a transporter gene onto results of Fig.2, to help elucidate the molecular mechanism of variation of starch property even partially.

Response

Discussion on the association of markers and clustered groups was added and the information about banding patterns was added to the Supplementary Table S7.

Minor comments:

  1. P2L93-95----“The value of the starch property (----) was evaluated among the 230 rice cultivars (Supplemented Table S1).” ----- TS1 shows only collective correlations among rice varieties and TS2 might be an appropriate table for this content.

Response

We switched TS1 and TS2 for the appropriate meaning as suggested.

  1. P11L240,242,243 marker----to be a molecular marker or DNA marker.

Response

It was changed to a molecular marker.

  1. P13L293,294---indica, japonica subspecies.

Response

These words were revised as suggested.

  1. P14L368---In the previous study,

Response

These words were revised as suggested.

  1. P15L416, P16L431---unify the description as “mixing at 200 rpm at 37°C for 60 min.”

Response

The method was revised for clarification.

  1. P17L470---reward primer---reverse primer?

Response

It should be a reverse primer. Thank you for pointing this.

  1. P18L539-----The authors express gratitude to Professor Luca Comai, but this makes a self-contradiction because he is one of the authors of this manuscript.

Response

We would like to thank Professor Luca Comai’s lab members. Therefore, we revised it for clarification.

Sincerely,

Supachitra Chadchawan

Reviewer 2 Report

Major Revisions:

1. Inclusion of RNA-seq Analysis and Gene Expression Analysis: Including RNA-seq analysis or gene expression analysis of selected candidate genes in the paper would enhance the overall research approach.

2. Strengthening of GWAS Analysis and Generation of Selection Markers Explanation: Adding detailed explanations of the GWAS analysis process and the results of selection marker generation would improve the validity and reliability of the research.

3. Addition of Statistical Analysis: Incorporating statistical analyses like Duncan's analysis into Figure 7 and Figure 8 would provide a higher level of confidence in the results.

4. Explanation for Exclusion of Selection Markers on Chromosome 1: Providing a rationale for not developing selection markers for chromosome 1, despite their presence on chromosomes 6 and 11, would help readers understand this decision better.

Minor Revisions:

 1. Correction of Candidate Gene Count in Table 1: The count of candidate genes associated with the AAC trait is listed as 5 in Table 1, but it is depicted as 6 in both Figure 5 and the manuscript. Consistency in this count should be maintained.

2. Correction of Candidate Gene Count for AAC and RS Traits: The count of candidate genes associated with both AAC and RS traits is stated as 12 in Table 1, but 11 in Figure 5. Ensuring consistency in this count is essential.

3. Adding Chromosome Number Beside SNP Position in Line 208: Including the chromosome number alongside the SNP position in Line 208 will enhance the visibility of the information.

4. Italic Formatting for Genes and Enzymes in Lines 242, 243, 247: Genes and enzymes should be written in italic form in lines 242, 243, and 247 to maintain the correct formatting.

Author Response

Response to Reviewer II

Thank you very much for your comments. I would like to response to your comments as follows.

Major Revisions:

  1. Inclusion of RNA-seq Analysis and Gene Expression Analysis: Including RNA-seq analysis or gene expression analysis of selected candidate genes in the paper would enhance the overall research approach.

Response

We agree with the reviewer that RNA-seq Analysis and Gene Expression Analysis would enhance the overall research approach. However, due to the time limitations, for this revision process, we could not complete this experiment. We will further our research with this suggested approach. Thank you so much for the suggestion.

  1. Strengthening of GWAS Analysis and Generation of Selection Markers Explanation: Adding detailed explanations of the GWAS analysis process and the results of selection marker generation would improve the validity and reliability of the research.

Response

The details of the GWAS and marker selection were added to the manuscript.

  1. Addition of Statistical Analysis: Incorporating statistical analyses like Duncan's analysis into Figure 7 and Figure 8 would provide a higher level of confidence in the results.

Response

The statistical analysis was performed and the added to the figures as suggested.

  1. Explanation for Exclusion of Selection Markers on Chromosome 1: Providing a rationale for not developing selection markers for chromosome 1, despite their presence on chromosomes 6 and 11, would help readers understand this decision better.

Response

Primers to detect the significant SNP on chromosome 1 could not be developed, because it was a single SNP, which could not be detected by this method. This information was added to Molecular marker analysis section.

Minor Revisions:

  1. Correction of Candidate Gene Count in Table 1: The count of candidate genes associated with the AAC trait is listed as 5 in Table 1, but it is depicted as 6 in both Figure 5 and the manuscript. Consistency in this count should be maintained.

Response

Thank you very much for pointing this to us. There was a mistake in the figure preparation, and we have corrected it.

  1. Correction of Candidate Gene Count for AAC and RS Traits: The count of candidate genes associated with both AAC and RS traits is stated as 12 in Table 1, but 11 in Figure 5. Ensuring consistency in this count is essential.

Response

Thank you very much for pointing this to us. We have to apologize for this mistake. There was a mistake in the figure preparation, and we have corrected it.

  1. Adding Chromosome Number Beside SNP Position in Line 208: Including the chromosome number alongside the SNP position in Line 208 will enhance the visibility of the information.

Response

We have added as your suggestion.

  1. Italic Formatting for Genes and Enzymes in Lines 242, 243, 247: Genes and enzymes should be written in italic form in lines 242, 243, and 247 to maintain the correct formatting.

Response

According to Gene Nomenclature System for Rice (McCouch et al., 2008), the gene name is written with italic letters, while the protein name is written with non-italic letters. We have revised the manuscript as they were indicated in McCouch et al. (2008)

McCouch, S. R. and CGSNL. 2008. Gene Nomenclature System for Rice. Rice. 1, 72–84.

Sincerely yours,

Supachitra Chadchawan

Reviewer 3 Report

A large and expensive study has been carried out revealing new insights into the genome regarding starch. The manuscript is characterized by a careful and consistent description that fully meets the requirements of the journal. And I have just a few observations:

I would suggest not to use abbreviations in the conclusions.

Looking at the data, a natural question arises - why were the varieties with extremely low amylose content (Wx) not included in the study? - I guess that this was taken into account when compiling the study. - Therefore, the setting of the lower values of the AM method raises doubts (of course - it does not have essential impact on the correlations).

Also, the description of the starch determination  method (from 410 line) should be clarified.  Please specify, from where the TG aliquot starts.... What amount of sample to be taken for G20, G120 and TG  concentrations estimation, since different dilutions are carried out. 

Author Response

Response to Reviewer III

Thank you very much for your comments and suggestions. I would like to respond to your comments as follows:

  1. I would suggest not to use abbreviations in the conclusions.

Response

We revised as recommended.

  1. Looking at the data, a natural question arises - why were the varieties with extremely low amylose content (Wx) not included in the study? - I guess that this was taken into account when compiling the study. - Therefore, the setting of the lower values of the AM method raises doubts (of course - it does not have essential impact on the correlations).

Response

The materials used in this research are mainly the local Thai rice. All of them are indica rice. Rice cultivars with various AAC were selected from the expert from rice Genebank of Thailand, Rice Department.

  1. Also, the description of the starch determination  method (from 410 line) should be clarified.  Please specify, from where the TG aliquot starts.... What amount of sample to be taken for G20, G120 and TG  concentrations estimation, since different dilutions are carried out. 

Response

The digestible starch measurement has been revised and clarified.

Sincerely yours,

Supachitra Chadchawan